# Seismic Experimental Assessment of Remote Terminal Unit System with Friction Pendulum under Triaxial Shake Table Tests

**Sung-Wan Kim** [1], **Bub-Gyu Jeon** [1], **Da-Woon Yun** [1], **Woo-Young Jung** [2,*] **and Bu-Seog Ju** [3,*]

1   Seismic Research and Test Center, Pusan National University, Yangsan-si 50612, Korea;
    swkim09@pusan.ac.kr (S.-W.K.); bkjeon79@pusan.ac.kr (B.-G.J.); ardw818@pusan.ac.kr (D.-W.Y.)
2   Department of Civil Engineering, Gangneung-Wonju National University, Gangneung-si 25457, Korea
3   Department of Civil Engineering, Kyung Hee University, Yongin-si 17104, Korea
*   Correspondence: woojung@gwnu.ac.kr (W.-Y.J.); bju2@khu.ac.kr (B.-S.J.)

**Abstract:** In recent years, earthquakes have caused more damage to nonstructural components, such as mechanical and electrical equipment and piping systems, than to structural components. In particular, among the nonstructural components, the electrical cabinet is an essential piece of equipment used to maintain the functionality of critical facilities such as nuclear and non-nuclear power plants. Therefore, damage to the electrical cabinet associated with the safety of the facility can lead to severe accidents related to loss-of-life and property damage. Consequently, the electrical cabinet system must be protected against strong ground motion. This paper presents an exploratory study of dynamic characteristics of seismically isolated remote terminal unit (RTU) cabinet system subjected to tri-axial shaking table, and also the shaking table test of the non-seismically isolated cabinet system was conducted to compare the vibration characteristics with the cabinet system installed with friction pendulum isolator device. In addition, for the shaking table test, two recorded earthquakes obtained from Korea and artificial earthquakes based on the common application of building seismic-resistant design standards as an input ground motions were applied. The experimental assessment showed that the various damage modes such as door opening, the fall of the wire mold, and damage to door lock occurred in the RTU panel fixed on the concrete foundation by a set anchor, but the damage occurred only at the seismic isolator in the seismically isolated RTU panel system. Furthermore, it was considered that the application of the seismic isolator can effectively mitigate the impact and amplification of seismic force to the RTU panel system during and after strong ground motions in this study.

**Keywords:** electrical cabinet; triaxial shake table; friction pendulum; seismic isolated device

## 1. Introduction

The frequency of earthquakes with a moment magnitude greater than 5.0 has been continuously increasing worldwide. Specifically, the frequency of earthquake occurrences on the Korean Peninsula has been significantly increasing since 1973. Magnitudes exceeding 4.0 occurred more than 50 times in Korea in 2019, and those with magnitudes over 5.0 occupied more than ten distributions. Consequently, the demand for a seismic design to mitigate social and economic losses caused by strong ground motion has begun to be emphasized. However, such seismic designs are upgraded and strengthened by focusing only on the structural components. Recently, nonstructural systems, such as mechanical and electrical equipment, suffered relatively extensive damage during strong ground motion, even when the primary system had no structural damage [1]. In addition, in moderate seismic zones, such as the Korean Peninsula, earthquake damage is concentrated on nonstructural components; nonstructural earthquake damage, including damage to electrical equipment and piping systems, was observed in the 2016 Gyeongju and 2017 Pohang earthquakes in Korea [2,3].

Electrical cabinets, a representative mechanical and electrical nonstructural component, are responsible for system control and communication to maintain essential functionality and operation in critical facilities. Damage to the electrical cabinet can cause a malfunction of critical facilities during and after strong earthquakes, and damage to the mechanical and electrical equipment in power plants can lead to large social and economic losses. Therefore, the structural safety of an electrical cabinet system subjected to strong ground motion is required; many studies have addressed the dynamic characteristics and seismic performance of electrical cabinets [4–7]. Furthermore, in-cabinet response spectra have been studied for estimating the dynamic characteristics of the instruments in an electrical cabinet system [8,9], and a study of rocking and amplification of floor acceleration due to impact has also been conducted for the electrical cabinet system [10,11]. It has been reported that damage to the locking device and support parts can occur, and structural damage, such as dropping of components in the cabinet, can also occur during strong earthquakes. Previous studies have confirmed that door shaking, rocking, and lifting amplified the in-cabinet response acceleration significantly [12].

For the case of Japan, to mitigate earthquake damage, vibration control technology, such as damper systems, has been applied for low-rise buildings, and seismically isolated systems have been utilized for high-rise buildings, rather than classical seismic design through reinforcement of structures. Typically, seismic isolation devices, such as natural rubber bearings (NRBs), high-damping rubber bearings (HDRBs), and friction pendulum systems (FPSs), are utilized for reducing the influence of seismic excitation on the structures. This is because these devices can increase the natural period of the superstructures owing to the installation between the bottom of the superstructure and the boundary area of the foundation during a strong ground motion [13]. In addition, the effect of seismic isolators on seismic load resisting systems has been proven over the past few decades, and these isolators have been applied to critical infrastructures, such as buildings and bridges [14]. Therefore, seismic isolation devices are also being applied to electrical equipment to secure safety from impacts or seismic wave vibration without complicated design changes. For example, a previous study considered the application of a seismic isolator to the emergency diesel generator used in nuclear power plants, and seismic fragility assessment was evaluated [15,16].

The characteristics of seismic behavior in electrical equipment with various seismic isolation devices have been experimentally explored [17,18]. The seismic safety of substation facilities using seismic isolators has also been studied [19–22]. To protect the electrical cabinet in critical facilities during and after earthquake ground motion, a study related to floor isolation systems corresponding to multidirectional spring units and FPSs was conducted [23,24]. The seismic isolation table and FPS based on springs and linear motion (M) guides as seismic isolators are commonly utilized to improve the seismic performance of broadcasting and communication devices and electrical equipment in Korea. The seismic performance of such systems can be estimated by shaking table tests based on ICC-ES AC 156 [25] and by the broadcasting and communication facilities seismic test method [26]. For the case of the friction pendulum system, the characteristics of vibration attenuation of electrical and mechanical equipment were analyzed by performing shaking table tests [27]. Most recent studies utilize a scale model experiment of a seismic isolation system, but there is a limitation in representing the on-site installation conditions. In addition, the scale model test was not able to adequately demonstrate the limitation of the asymmetric center-of-gravity of the electrical cabinet system by simulating the load of the device. The use of triaxial shaking table tests, considering transverse, longitudinal, and vertical directions for the three directional vibration characteristics and attenuation of the seismically isolated system, is also limited. Based on the Seismic Building Design Code and Commentary (Korean Building Code) [28], essential non-structural components, such as mechanical and electrical equipment, that must remain functional during and after strong ground motion are required to prove their seismic performance through a shaking table test. ASCE 7 [29] reported that mechanical and electrical equipment must satisfy the seismic load and rel-

ative displacement requirements, and that no functional electrical defects should occur. With respect to such non-structural components, the required seismic performance can be verified by performing tests such as ICC-ES AC 156.

This paper presents the characteristics of vibration transmissibility of seismically isolated electrical cabinet systems associated with ICC-ES AC 156 using a triaxial shaking table. First, the electrical cabinet is anchored to a concrete slab on a shaking table. Next, to compare the vibration characteristics of the cabinet system anchored on the concrete slab, an electrical cabinet is installed using an FPS comprising a stopper and spring system. The stopper in the FPS can resist overturning, and the spring system prevents the separation of the upper structure and improves the additional restoring force. Finally, the evaluation of the vibration characteristics of cabinet systems based on the seismic building design code are analyzed. Artificial earthquake motion is generated by the required response spectra according to ICC-ES AC156. The triaxial shaking table tests, consisting of a combination of two horizontal and one vertical direction time histories, are conducted for two different types of electrical cabinet systems to overcome the limitation of the test results. Once the vibration characteristics under various types of ground motion, including artificial and recorded earthquakes, are determined, the behavior of the anchored electrical cabinet system on the concrete slab is compared with that of a seismically isolated electrical cabinet using an FPS.

## 2. Friction Pendulum System

The FPS, a seismic isolator of the friction series, is a device that can determine the natural frequency of the target structure by using the characteristics of the pendulum. As shown in Figure 1, it moves like a pendulum along a horizontal direction on a curved surface to show the effect of seismic isolation based on the dissipation of seismic energy. The horizontal direction force (*F*) of the FPS can be expressed as

$$F = \frac{W}{R}U + \mu W \sin(\dot{u}) \tag{1}$$

where *W*, *R*, *U*, *μ*, and *u̇* represent the vertical load from the upper structure, radius of curvature of the curved surface, horizontal displacement, surface friction coefficient, and velocity, respectively. The natural period of the FPS can be determined by *R*, regardless of the upper structure mass, according to Equation (2).

$$T = 2\pi\sqrt{\frac{R}{g}} \tag{2}$$

where *g* represents the acceleration due to gravity. In addition, the natural frequency (Hz) can be transformed by the natural period (*T*), as described in Equation (3), and the natural frequency of the FPS targeted 1.0 Hz in this study.

$$f = \frac{1}{T} \tag{3}$$

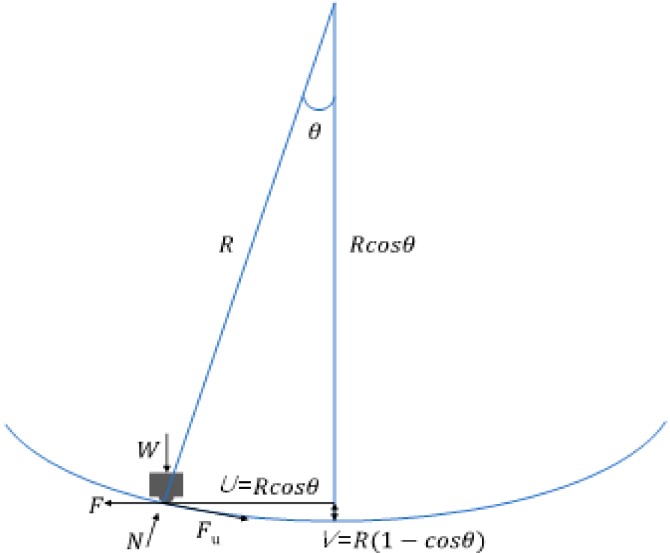

**Figure 1.** Typical principle of a friction pendulum system.

### 3. Target System and Installation Conditions

A remote terminal unit (RTU) is a device that collects data from remote locations and transmits them to a higher-level target system. In addition, the RTU, an important device in power plants, can perform the procedure to control the power generation facility from the received data. Therefore, this study selected a group of existing RTU systems in a hydroelectric power plant and then conducted a tri-axis shaking table test to obtain reliable test results. Table 1 presents the details of the targeted RTU panel and seismic isolator. Regarding the existing RTU panel, the mass of the grouped cabinet system is illustrated, and the mass of the seismic isolator devices include a jig to fix the shaking table.

**Table 1.** Description of units under test.

| Name | Model (Detail) | Specifications [mm] | | | Mass [kg] |
| :---: | :---: | :---: | :---: | :---: | :---: |
| | | Length | Width | Height | |
| RTU panel | RTU No.2-1 | 880 | 620 | 2350 | 518 |
| | RTU No.3-1 | 880 | 620 | 2350 | |
| Isolation system | Seismic isolation system (Ball transfer type FPS with non-overturning stopper) | 1750 | 860 | 110 | 373 |

The existing FPS can cause a separation of the ball and impact due to uplift induced by vertical vibration, and can cause overturning of the cabinet equipment due to unstable conditions. The seismic isolator, used as a type of FPS in this study, is a non-overturn seismic isolator manufactured by Power&Tech Co., Ltd, Seongnam-si, Korea. To prevent the separation of the ball, a ball transfer, as shown in Figure 2a, was installed, and the spring connecting the upper and lower plates of the seismic isolator system ensures horizontal and vertical restoring forces, as shown in Figure 2b. Figure 2c illustrates the system with a spring as a type of FPS.

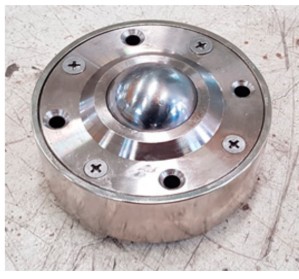
(**a**) Ball transfer

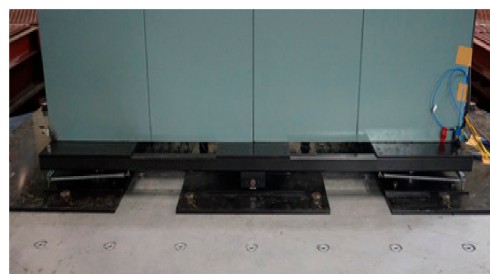
(**b**) RTU on the FPS

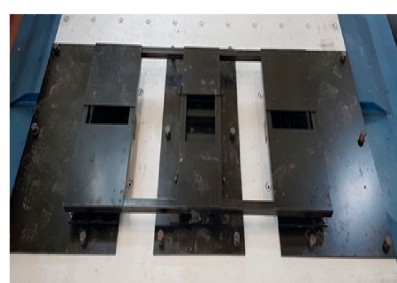
(**c**) Spring

**Figure 2.** Non-overturn seismic isolator on the shaking table.

Next, a triaxial shake table test corresponding to the conditions listed in Table 2 was conducted. Case 1 was applied to illustrate the on-site installation conditions of the RTU system. After fixing the enclosure of the grouped RTU panel to one base channel, as shown in Figure 3a, the RTU system was installed using a set anchor on the concrete foundation fixed to the shaking table. The RTU panel was connected to the base channel using eight M8 bolts. Both ends of the base channel were fixed to the concrete foundation with two M12 set anchors, and two L-shaped angles in front of the RTU panel were connected to the base channel by welding. Next, it was fixed to the concrete with each M8 set anchor. For Case 2, as described in Figure 3b, the RTU was fastened to the top plate of the seismic isolator using eight M8 bolts, and then the seismically isolated device was coupled with M30 bolts on the shaking table. In addition, the design properties in terms of M8 and M12 set anchors listed in the Table 3, and the tensile resistance force of M30 set anchors as a high strength bolt was 403 kN.

**Table 2.** Detailed description of shaking table tests.

| Cases | Mounting Description | |
|---|---|---|
| | **Shake Table—Base of UUT** | **Base of UUT—Bottom Plate of RTU Panel** |
| Case 1 | Base channel anchored using set anchor on the concrete slab | RTU panel fixed on the base channel using bolts |
| Case 2 | Seismic isolation system | RTU panel fixed on the isolation system using bolts |

UUT: unit under test.

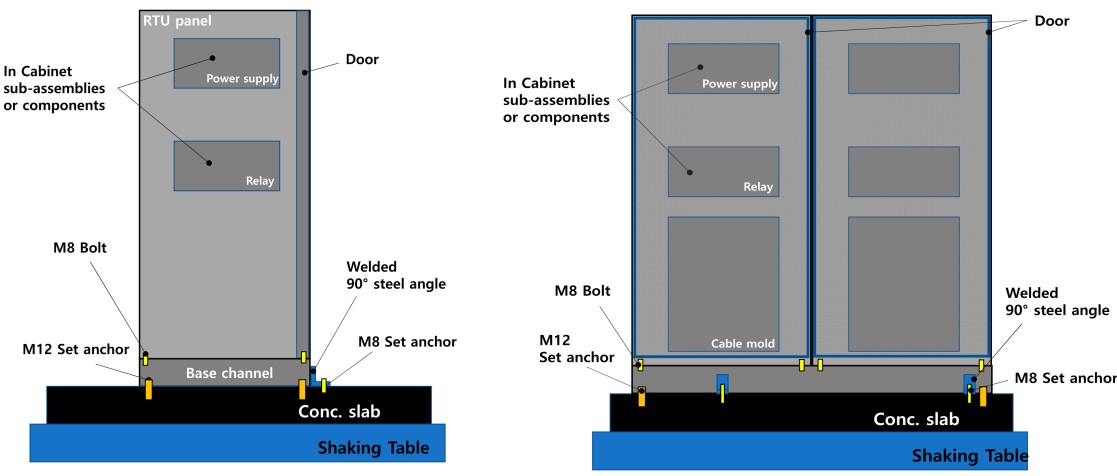
(**a**) Case 1: RTU panel on concrete slab

**Figure 3.** *Cont.*

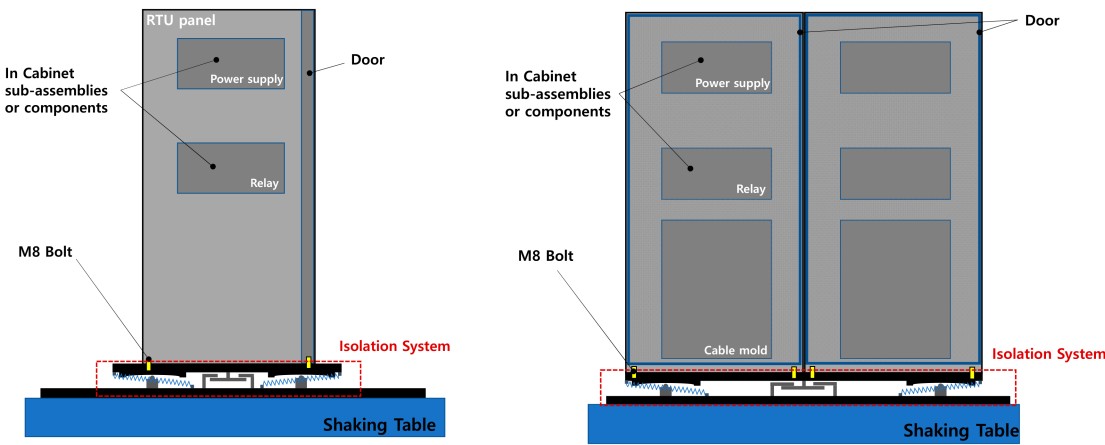

(**b**) Case 2: Base-isolated RTU panel

**Figure 3.** Description of the RTU systems on the shaking table.

**Table 3.** Description of the set anchors M8 and M12.

| Size | Overall Length (mm) | Screw Length (mm) | Diameter (mm) | Tensile Force (kN) |
|------|---------------------|-------------------|---------------|--------------------|
| M8   | 60                  | 40                | 12            | 18                 |
| M12  | 150                 | 100               | 17            | 31                 |

## 4. Target Input Ground Motion

### 4.1. Artificial Earthquakes

In this study, to consider both cases where electrical cabinets were installed in general buildings and industrial facilities, the common application of seismic design criteria in Korea was considered [30]. A required response spectrum (RRS) corresponding to ICC-ES AC 156 [31] was generated accordingly. The design spectral response acceleration $S_{DS}$ for a short period from the seismic design code of buildings in Korea can be obtained using Equation (4).

$$S_{DS} = S \times 2.5 \times f_a \times \frac{2}{3} \tag{4}$$

where $S$ and $f_a$ denote the effective ground acceleration and amplification factor associated with short-period site conditions, respectively. In Equation (5), $S$ was determined to be 0.22, with the coefficient of seismic zone ($Z$) and the risk factor ($I$) based on a 2400-year return period. In this case, $f_a$ was assumed to be 1.5, and the calculated $S_{DS}$ was 0.55. The calculation method of $S$ for the common application of seismic design criteria was identical to that of the seismic design code of buildings.

$$S = Z \times I \tag{5}$$

$$S_{DS} = S \times 2.5 \times f_a \tag{6}$$

However, $S_{DS}$ in the common application of seismic design criteria was more conservative than that of the seismic design code of buildings, as expressed by Equation (6). Equations (5) and (6) show that the design spectral response acceleration for a short period of the common application of seismic design criteria is 1/3 times larger than that of seismic design code of buildings. For instance, if $f_a$ is 1.5 and $S$ is 0.22, $S_{DS}$ is 0.55 (seismic design code of buildings) and 0.825 (common application of seismic design criteria). In this study, the acceleration magnifications were defined as: (1) EQ #1 (100% acceleration with 0.55 of $S_{DS}$) and (2) EQ #2 (150% acceleration with 0.825 of $S_{DS}$). Next, the acceleration magnification was amplified by increments of 50% (EQ #3 and EQ #4), and the test was

performed until structural or functional damage occurred. Figure 4a demonstrates the RRS based on ICC-ES AC 156 considering the floor level response of the structural system, and Figure 4b shows the time histories of each direction of EQ1. Table 4 lists the parameters used to generate the RRS.

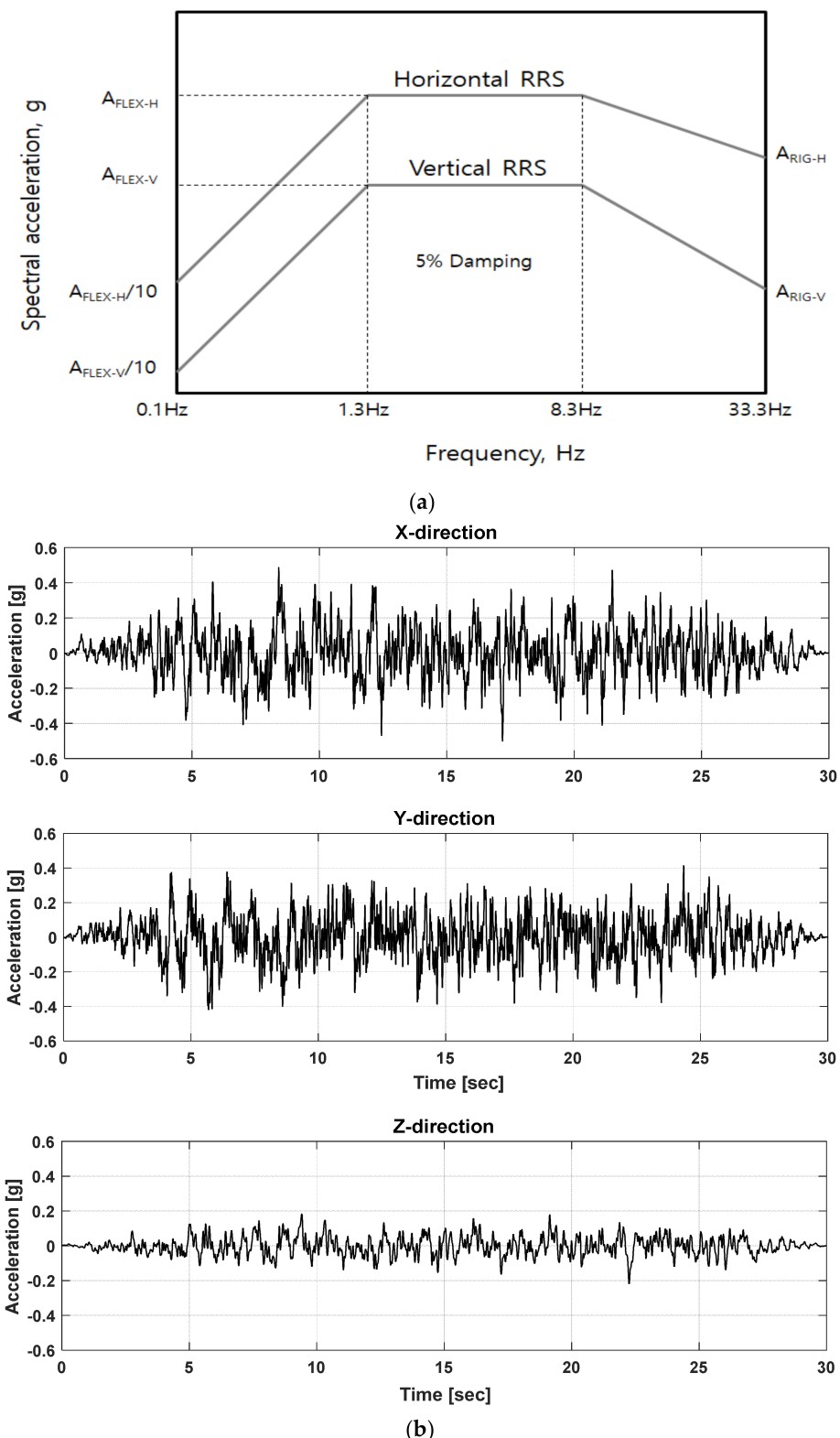

**Figure 4.** Input motion (**a**) RRS based on ICC—ES AC 156, (**b**) acceleration time histories.

**Table 4.** Parameters used to generate RRS.

| Earthquakes | Amplification [%] | Code | $S_{DS}$ [g] | $z/h$ | $A_{FLEX\text{-}H}$ [g] | $A_{RIG\text{-}H}$ [g] | $A_{FLEX\text{-}V}$ [g] | $A_{RIG\text{-}V}$ [g] |
|---|---|---|---|---|---|---|---|---|
| EQ #1 | 100 | Seismic design code of buildings | 0.55 | 1 | 0.88 | 0.66 | 0.36 | 0.14 |
| EQ #2 | 150 | Common application of seismic design criteria | 0.825 | 1 | 1.32 | 0.99 | 0.55 | 0.22 |
| EQ #3 | 200 | - | 1.10 | 1 | 1.76 | 1.32 | 0.73 | 0.29 |
| EQ #4 | 250 | - | 1.375 | 1 | 2.20 | 1.65 | 0.92 | 0.37 |

Electrical equipment, such as cabinet systems, can be installed on all floor levels in critical facilities. Accordingly, the ratio of height in structure of level of cabinet component in terms of base ($z$) and average roof height of the facilities in terms of base can be determined as $z/h = 1$, which is a conservative approach. Moreover, we assumed that the damping ratio of the RRS is 5%. Therefore, the horizontal spectral acceleration of the RRS can be determined using Equation (7) based on the flexible component ($A_{FLX}$), and Equation (8) based on the rigid component ($A_{RIG}$). Similarly, Equations (9) and (10) describe the characterization of vertical components [31,32]. At this point, $A_{FLEX\text{-}H}$ corresponding to ICC-ES AC156 cannot exceed 1.6 times the design spectral response acceleration in a short period.

$$A_{FLEX-H} = S_{DS}\left(1 + 2\frac{z}{h}\right) \tag{7}$$

$$A_{RIG-H} = 0.4 S_{DS}\left(1 + 2\frac{z}{h}\right) \tag{8}$$

$$A_{FLEX-V} = 0.67 S_{DS} \tag{9}$$

$$A_{RIG-V} = 0.27 S_{DS} \tag{10}$$

The acceleration time history was implemented using a trapezoidal envelope function in accordance with ASCE 4-98 [33], and the correlation of each axial direction ($x$, $y$, $z$) was examined. The coefficient of correlation function at $xy$, $xz$, and $yz$ planes was set as 0.3 or less, based on ASCE 4-98 and IEEE Std. 344 [34]. The duration of the acceleration time history with respect to the artificial ground motion was 30 s and, in this case, the duration of the strong ground motion was 20 s. Furthermore, with this acceleration, the magnitude of the acceleration increased for the first 5 s, and then the strong ground motion continued for 20 s. Subsequently, the magnitude of acceleration decreased during the last 5 s.

*4.2. Recorded Earthquakes*

In this study, among historical earthquakes in Korea, the Gyeongju and Pohang earthquakes, which caused significant damage, were selected as input ground motions to conduct the shaking table test. The peak ground acceleration (PGA) of the Gyeongju earthquake (USN) and Pohang earthquake (PHA2) was 0.43 g and 0.27 g, respectively. In addition, each ground motion exhibited different frequency components. Overall, the spectral acceleration of the Pohang earthquake was greater than that of the Gyeongju earthquake in the low-frequency range (below 2 Hz). However, the Gyeongju earthquake showed greater spectral acceleration than the Pohang earthquake in the high-frequency range (over 10 Hz). Figure 5 demonstrates the acceleration time history and response spectra for Gyeongju and Pohang.

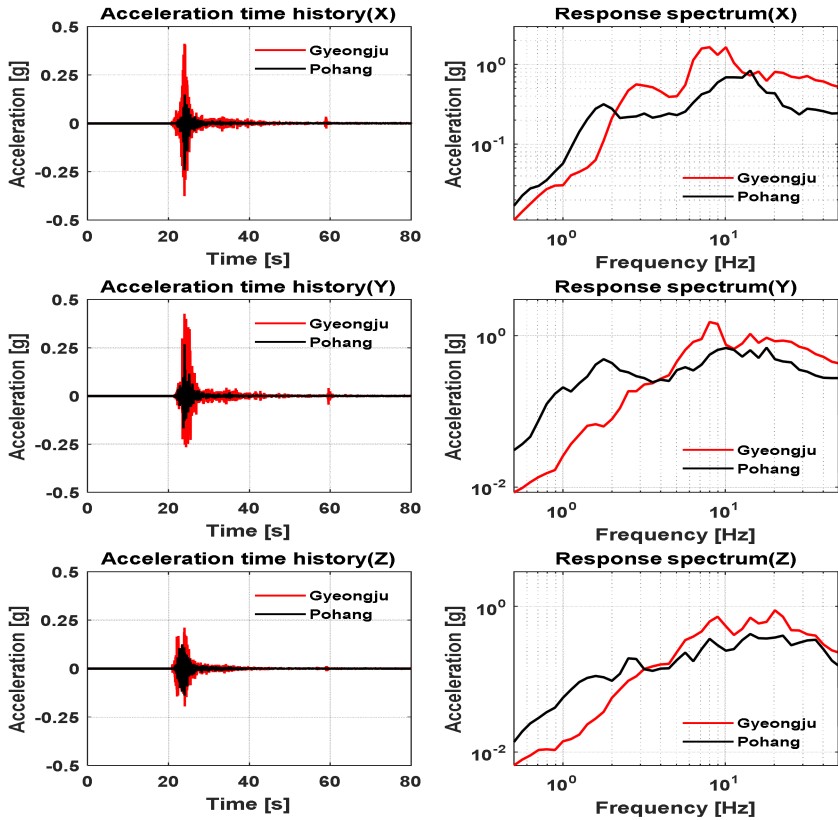

**Figure 5.** Acceleration time histories and response spectra for Gyeongju (USN) and Pohang (PHA2) earthquakes.

## 5. Shaking Table Tests

### 5.1. Sensing

In this experimental test, the acceleration response was measured by installing three-axis accelerometers (A2–A6) around the bottom, center, and top of the RTU panel, and the response was obtained around the power supply and relay of the internal main parts of the RTU. Next, to achieve a reliable test response, a three-axis accelerometer (A1) was placed at the bottom of the shaking table. A strain gauge was installed in the horizontal direction, perpendicular to the position 20 mm away from the center of the bolt (SG 1–SG 8). This is because, due to the excessive external force, there is a risk of local deformation at the bottom plate of 2-mm thickness around the bolt connecting the RTU panel and base channel. The strain gauge was set up around the four connecting bolts at the front, as shown in Figure 6, and the displacement sensors were installed at the top and bottom to measure the relative displacement of the RTU panel caused by the seismic input motion. For Case 1, only a load cell was installed to measure the pull-out force of the post-installed anchor connected to the concrete slab and lower channel. Before, during, and after the test, the electrical signal of the main circuit was measured using an AC/DC transducer to monitor the functionality. Figure 6a,b illustrate the location of the sensors according to cases 1 and 2, respectively, and Figure 6c shows the schematic design of the stain gauges.

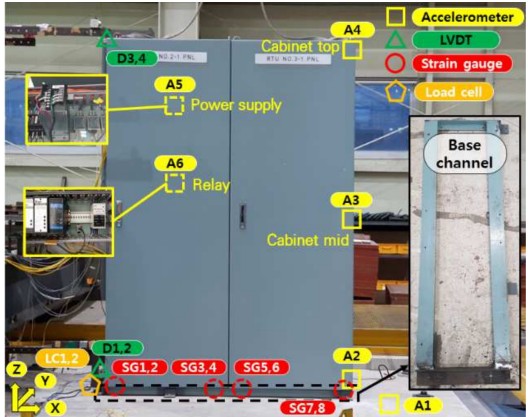

(**a**) Sensor location for Case 1

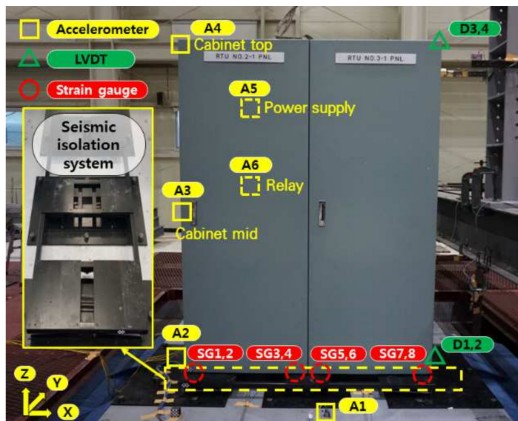

(**b**) Sensor location for Case 2

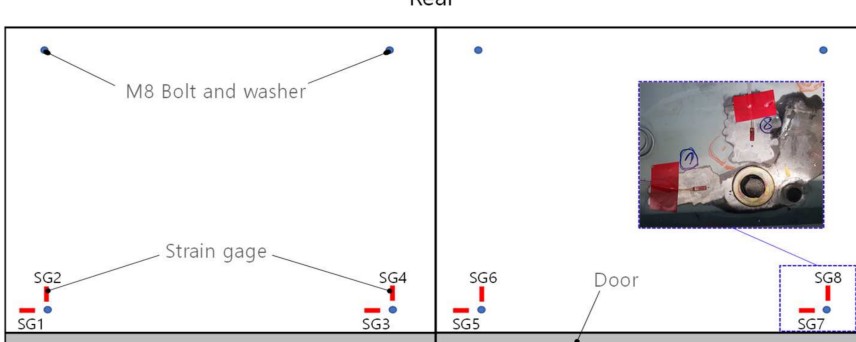

(**c**) Schematic design of strain gauges on the bottom plate

**Figure 6.** Sensor locations for the cabinet.

### 5.2. Experimental Test Procedure

The test procedure was performed using ICC-ES AC 156, as listed in Table 5, and Table 6 presents the test sequence.

**Table 5.** Test configuration.

| No. | Test Name | Test Method |
|-----|-----------|-------------|
| 1 | Visual inspection | - |
| 2 | Function verification | on-off-on test, voltage signal measurement |
| 3 | Resonant frequency search | low-level amplitude (0.05 g)<br>single-axis sinusoidal sweep (0.5–50.0 Hz),<br>2 octave/min<br>X, Y, Z axis independently |
| 4 | Seismic simulation<br>(Recorded earthquakes) | Gyeongju earthquake<br>Pohang earthquake |
| 5 | Seismic simulation<br>(Artificial earthquake) | AC 156 100%–250%, time duration 30 s,<br>Strong motion 20 s, damping ratio 5%, 0.5–50 Hz,<br>triaxial test |

Before and after all the tests, the power was turned on and off to verify the integrity of the circuit, and the dropped parts and deformation of the system were investigated through visual inspection. To confirm the resonant frequency of the RTU system, a single-axis sinusoidal sweep was conducted in each direction, such as the rear–front, left–right, and up–down, with a low input acceleration (0.05 g). The seismic simulation test consisted

of two horizontal axes (front and rear, left and right) and one vertical axis (up and down), and it was performed such that the test response spectrum (TRS) measured at the bottom of the shaking table was enveloped by the RRS.

**Table 6.** Test procedure of the shaking table.

| Test | Test Name | Earthquake Name |
|:---:|:---:|:---:|
| 1 | Resonant frequency search #1 | - |
| 2 | Seismic simulation (recorded earthquakes) #1 | Gyeongju earthquake |
| 3 | Seismic simulation (recorded earthquakes) #2 | Pohang earthquake |
| 4 | Resonant frequency search #2 | - |
| 5 | Seismic simulation (artificial earthquake) #1 | EQ #1 (100%) |
| 6 | Seismic simulation (artificial earthquake) #2 | EQ #2 (150%) |
| 7 | Seismic simulation (artificial earthquake) #3 | EQ #3 (200%) |
| 8 | Seismic simulation (artificial earthquake) #4 | EQ #4 (250%) |

## 6. Vibration Characteristics of Seismically Isolated RTU System

### 6.1. Resonant Frequency

The resonance of the RTU system was determined by computing the transfer function of the response acceleration (unit, b) at each position of the RTU to the input acceleration (base, a) from the shaking table during the resonance search experimental test. The transfer function ($T_{ab}$) can be calculated using the cross power spectral density ($P_{ba}$) of the input and output signals for the power spectral density ($P_{aa}$) of the input signal, as expressed by Equation (11).

$$T_{ab}(f) = \frac{P_{ba}(f)}{P_{aa}(f)} \tag{11}$$

The symmetric Hamming window was applied to each signal to achieve a reliable resonance analysis. Table 7 lists the experimental data corresponding to the resonance search tests. The lowest resonant frequency in the front–rear and left–right directions for Case 1 (non-isolated RTU system) was 5.25 Hz. However, the resonance frequency was 4.75 and 4.50 Hz after the completion of test 2 (GyeongJu earthquake) and 3 (Pohang earthquake), respectively. We observed that the change in the resonant frequency resulted in a structural change in the RTU panel. According to IEEE std. 693-2018 [35], which is a test method for verification of the design of substation facilities, structural damage of the system can occur if the change in the resonant frequency exceeds 20%. From our observation of this experimental test, the range of the change in the resonant frequency was approximately 14%. Furthermore, there was a trivial structural problem, regardless of slight yielding and loosening of bolts in the RTU system by visual investigation, during the Gyeongju and Pohang earthquakes. In addition, the resonant frequency in accordance with the vertical direction was not observed within the test frequency range.

In both cases 1 and 2 (seismically isolated RTU system), the resonant frequency observed at the internal relay position and that measured at the panel enclosure were similar. Consequently, for the case of the RTU panel, the characteristics of the dynamic vibration of the panel enclosure were transmitted to small parts, such as relays. However, for the case of a power supply, a heavy internal device fixed by a cantilever type by bolts, the resonant frequency in the front–rear and left–right directions exhibited the same frequency as that of the enclosure, but the resonant frequency in the vertical direction was significantly different. The vertical resonant frequency of the power supply was 12 Hz. After tests 2 and 3 in Case 2, it was 11.25 Hz, which was approximately 6.25% lower than that of the power supply. Furthermore, there was no change in the resonant frequency after tests 2 and 3 for Case 2, and the lowest resonant frequency measured in the resonance search test of Case 2 was 1.125 Hz in both the front–rear and left–right directions. Consequently, the

seismic isolated device significantly controlled the dynamic vibration of the RTU panel, and the resonant frequency in the vertical direction changed from 22.75 Hz to 21. 88 Hz; however, the slight gap was less than 4%.

**Table 7.** Estimated resonant frequencies from the experimental tests.

| Sensor Location | Dir. | Lowest Resonant Frequency (Hz) | | | |
|---|---|---|---|---|---|
| | | Case 1 | | Case 2 | |
| | | #1 | #2 | #1 | #2 |
| RTU panel bottom left (A2) | Side-to-side (X) | 15 | 15 | 1.125 | 1.125 |
| | Front-to-back (Y) | 15 | 15 | 1.125 | 1.125 |
| | Vertical (Z) | N/A | N/A | 22.75 | 21.88 |
| RTU panel middle left (A3) | Side-to-side (X) | 5.25 Hz | 4.50 Hz | 1.125 | 1.125 |
| | Front-to-back (Y) | 5.25 Hz | 4.75 Hz | 1.125 | 1.125 |
| | Vertical (Z) | N/A | N/A | 22.75 | 21.88 |
| Top of cabinet (A4) | Side-to-side (X) | 5.25 Hz | 4.75 Hz | 1.125 | 1.125 |
| | Front-to-back (Y) | 5.25 Hz | 4.75 Hz | 1.125 | 1.125 |
| | Vertical (Z) | N/A | N/A | 22.75 | 21.88 |
| Power supply (A5) | Side-to-side (X) | 5.25 Hz | 4.75 Hz | 1.125 | 1.125 |
| | Front-to-back (Y) | 5.25 Hz | 4.75 Hz | 1.125 | 1.125 |
| | Vertical (Z) | 12.00 Hz | 12.00 Hz | 12.0 | 11.25 |
| Near relay (A6) | Side-to-side (X) | 5.25 Hz | 4.50 Hz | 1.125 | 1.125 |
| | Front-to-back (Y) | 5.25 Hz | 4.75 Hz | 1.125 | 1.125 |
| | Vertical (Z) | N/A | N/A | 22.75 | 21.88 |

*6.2. Damage Observation*

After the shaking table tests, the power was turned on–off–on to confirm the functionality of the RTU system, and then a visual inspection was conducted to examine the parts for drop-off, deformation, and loosening of bolts. Figure 7 and Table 8 present the visual inspection results. For Case 1, the cable mold was disconnected during the Pohang earthquake, but there was no damage by visual inspection during the Gyeongju and Pohang earthquakes for Case 2. In the case of the seismic simulation test by an artificial earthquake, the cable mold was also uncoupled during Test 6 (EQ #2), and the fixed door device was damaged in Test 7 (EQ #3). Therefore, the door was opened during the test, and the test was terminated. After the shaking table tests, a cup-like deformation occurred around the bolt connection area of the lower plate at the RTU panel, and the M8 anchor set was also damaged, as shown in Figure 8.

The RTU system with the seismically isolated device, however, remained functional and operational during and after the EQ #3 (test 7) loading condition for Case 2. There was also no structural damage during EQ #4 (test 8), but the test was stopped because the fall prevention stopper of the seismic isolator was split, as illustrated in Figure 9. The fall prevention stopper was damaged by the impact generated by the friction pendulum, exceeding the allowable displacement caused by the seismic ground motion. Therefore, the allowable displacement of the slider plate of the friction pendulum system could be adjusted to sufficiently respond to the strong ground motion. Accordingly, we assumed that the uncertainty with respect to other structural damage in the RTU system could be reduced by concentrating the damage mode to the seismic isolators. Notably, the integrity of the electric circuit was maintained until all the seismic simulation tests were completed.

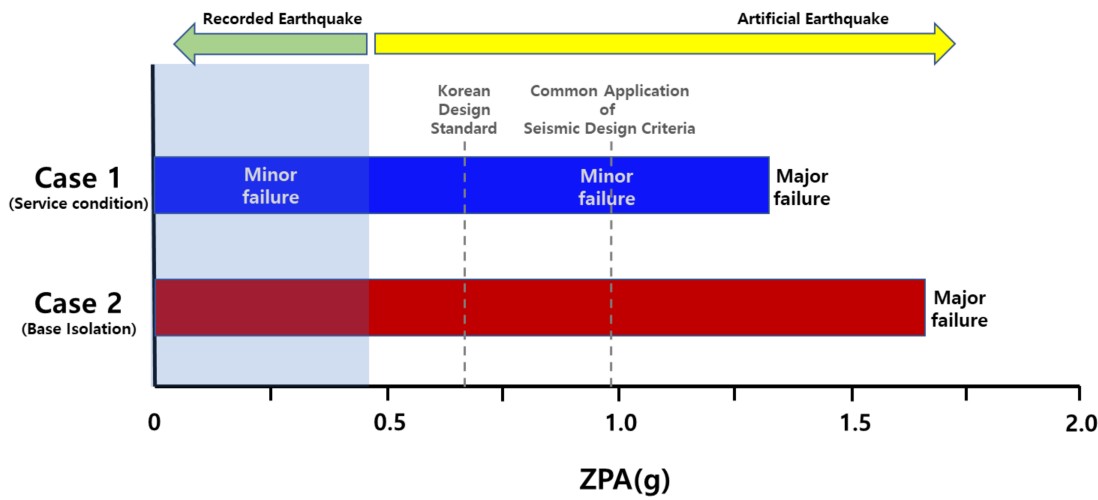

**Figure 7.** Visual inspection modes.

**Table 8.** Visual inspection observation.

| Test Name | | Visual Inspection | |
| --- | --- | --- | --- |
| | | Case 1 | Case 2 |
| Seismic simulation (recorded earthquakes) | #1 (Gyeongju earthquake) | Not found | Not found |
| | #2 (Pohang earthquake) | Cable mold separation | Not found |
| Seismic simulation (artificial earthquake) | #1 (EQ #1) | Not found | Not found |
| | #2 (EQ #2) | Cable mold separation | Not found |
| | #3 (EQ #3) | Door open Door lock failure M8 Set anchor failure | Not found |
| | #4 (EQ #4) | N/A | Stopper failure |

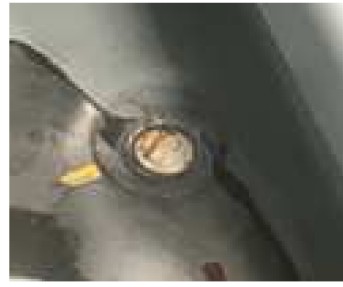

(**a**) Cup-like deformation

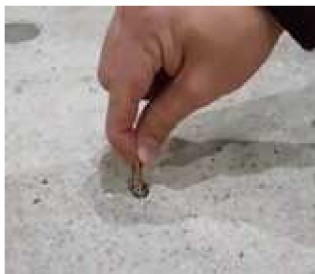

(**b**) M8 anchor set failure

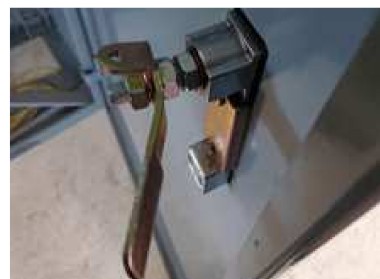

(**c**) Door-lock failure

**Figure 8.** Failure of the RTU system due to EQ #3 for Case 1.

*6.3. Response of Strain and Relative Displacement*

Figure 10a compares the strain response measured at the same location of the lower plate of the RTU panel for cases 1 and 2, and the relative displacement of the upper and lower parts of the RTU panel is described in Figure 10b. Specifically, the strain response of SG7 with relatively significant differences observed in the strain gauge around the fixing bolt, located on the lower plate of the RTU panel, was shown according to the test excitation sequence (Figure 10a). The black and red lines indicate responses of cases 1 and

2, respectively. In all seismic simulation tests, the strain response of Case 1 was significantly greater than that of Case 2. In addition, for Case 1, permanent strain occurred at EQ #1, and the value of 0.002 for the maximum strain response was exceeded at EQ #2. Finally, the permanent strain of 0.00396 occurred in EQ #3. However, for Case 2, the permanent strain of the system without the damage to the seismic isolator was less than 0.0001 up to EQ #3, and the permanent deformation was not as high as 0.00027 or less when the seismic isolator was damaged during the EQ #4 test. Therefore, when a seismic isolator was installed, the structural damage caused by the earthquake was concentrated on it, and the upper device was safely protected. The results of the shaking table test indicate that the deformation of the lower panel in the RTU can be effectively mitigated by the installation of the seismic isolator, with over 99% maximum reduction rate. Table 9 lists the permanent deformation with respect to EQ #3 and #4; herein, SG 1 and SG 8 were excluded due to the damage during the test.

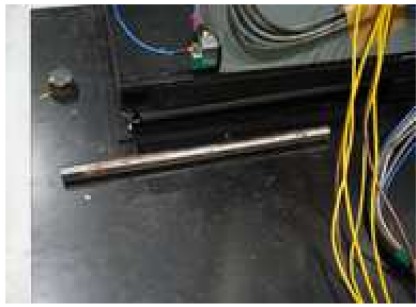
(**a**) Stopper failure

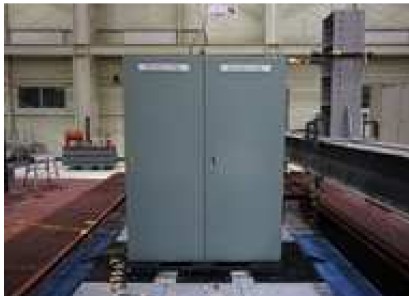
(**b**) RTU panel after the test

**Figure 9.** Failure of the RTU system due to EQ #4 (test 8) for Case 2.

**Table 9.** Permanent strain responses.

| | Sensor Location | Case1 ($\mu\varepsilon$) | Case2 ($\mu\varepsilon$) | | Reduction Ratio of EQ3 (%) |
|---|---|---|---|---|---|
| | | EQ #3 | EQ #3 | EQ #4 | |
| Permanent strain | SG2 | 500 | 10 | 80 | 98 |
| | SG3 | 380 | 80 | 40 | 80 |
| | SG4 | 250 | 90 | 270 | 65 |
| | SG5 | 410 | 100 | 140 | 75 |
| | SG6 | 250 | 100 | 250 | 60 |
| | SG7 | 3960 | 50 | 140 | 99 |

Table 10 and Figure 10b show the unidirectional maximum relative displacements in the left–right (X) and front–back (Y) directions for the top and bottom of the RTU panel. The relative displacement was compared up to EQ #3, in accordance with the results obtained from the experimental tests. In Figure 10b, in terms of the relative displacement time history responses, a signal capable of causing an impact was measured in all the tests of Case 1, wherein the base channel is fixed to a concrete slab with a post-installed anchor. In particular, Figure 10b depicts that the permanent deformation in Figure 10a occurs when a large instantaneous displacement with a short duration is generated by an impact.

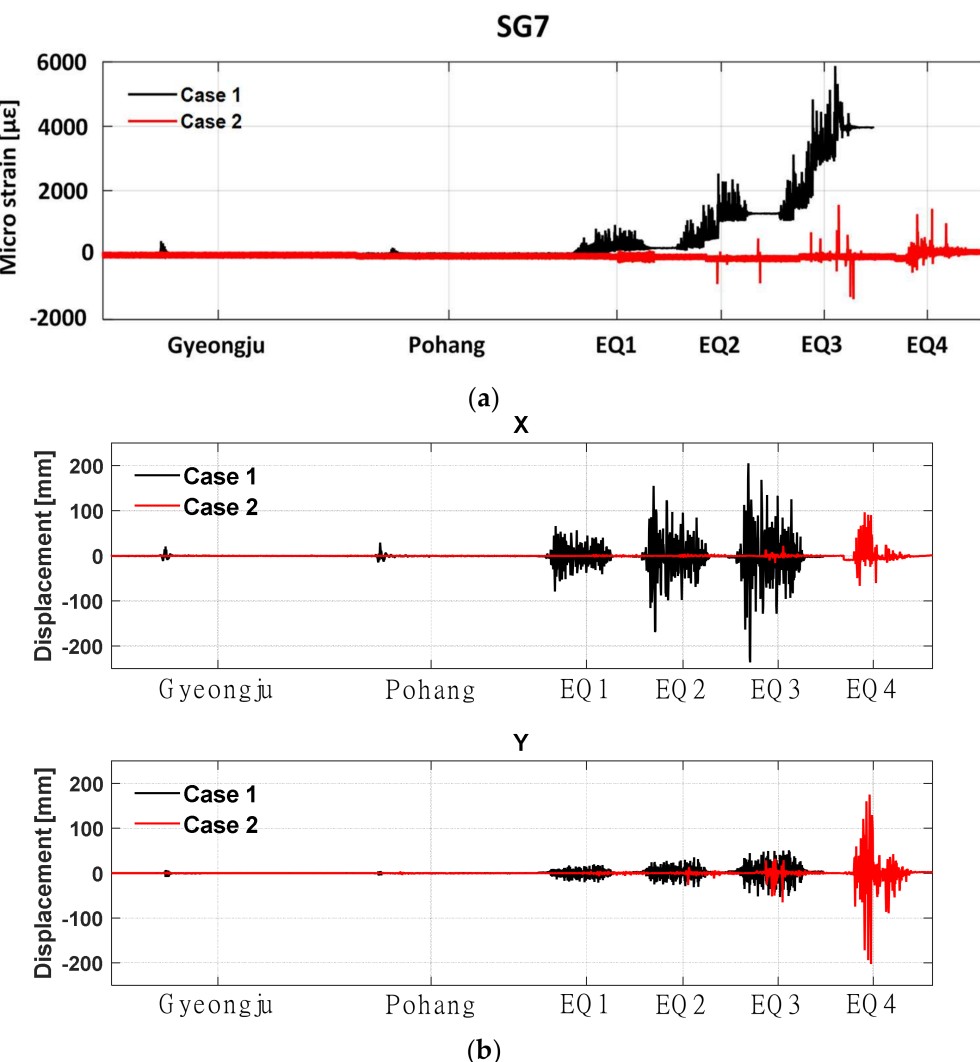

**Figure 10.** Response of strain and relative displacement of the RTU: (**a**) strain response and (**b**) di—placement response.

**Table 10.** Relative displacements under seismic simulation tests.

| Test No. | Earthquake | Case 1 (mm) | | Case 2 (mm) | | Case 1–Case 2 (mm) | | Reduction Ratio (%) | |
|---|---|---|---|---|---|---|---|---|---|
| | | X | Y | X | Y | X | Y | X | Y |
| 2 | Gyeongju | 21 | 9 | 1 | 3 | 20 | 6 | 95 | 67 |
| 3 | Pohang | 30 | 5 | 1 | 4 | 29 | 1 | 97 | 20 |
| 5 | EQ1 | 78 | 21 | 3 | 7 | 75 | 14 | 96 | 67 |
| 6 | EQ2 | 169 | 35 | 5 | 26 | 164 | 9 | 97 | 26 |
| 7 | EQ3 | 236 | 53 | 22 | 65 | 214 | −12 | 91 | −23 |

Such an impact not only caused structural damage to the cabinet, but also generated a high possibility of causing internal parts to fall off, or chattering. Therefore, the impact on the cabinet must be mitigated.

Uniaxial maximum relative displacement was observed in EQ #3 in both cases. The maximum relative displacements in the left and right directions (*X*) of Case 1 were 236 mm, and that in the front and rear directions (*Y*) was 53 mm. However, the uniaxial maximum relative displacement in Case 2 installed with the seismic isolator was 22 mm in the left and right directions (*X*), and 65 mm in the front and rear directions (*Y*). The relative displacement in the left and right directions (X) of the RTU applied to the seismic isolator

was reduced by more than 91%, in comparison to Case 1, as shown in Table 10. The relative displacement in the front and rear directions was reduced by at least 20% and at most by 67% up to EQ #2, in comparison to Case 1. For EQ #3, the displacement was increased up to 23%, whereas the relative displacement in the left and right directions was mitigated by 214 mm, in comparison to Case 1. However, the gap in the increased displacement with respect to the front and rear directions was only 12 mm in EQ #3. The Telcordia GR-63 core R4-69 [36] and the broadcasting and communication facilities seismic test method suggested a maximum allowable width displacement of 75 mm in one direction at the upper end of the cabinet system. For Case 2, the relative displacement measured in both horizontal directions up to EQ #3 was within 75 mm before the seismic isolator was damaged. In particular, in the Y-axis direction, the unidirectional relative displacement increased by 23% after the application of the seismic isolator, but the maximum relative displacement was within 75 mm. Therefore, no structural damage occurred.

Next, the magnitude of the strain was reduced by up to 90% and 99% in EQ #2 and #3, respectively, as shown in Figure 11, with respect to the permanent strain ratio (Case 2/Case 1). Specifically, the reduction in the strain response was significantly higher in SG 2 and SG 7, which were the strain gauges installed to measure the strain around the bolts, located at both ends of the lower panel of the RTU panel. Therefore, the end of the panel was lifted from shaking due to strong ground motion. This must be controlled because the uplifted cabinet system can cause a strong impact between the lower channel and cabinet. Finally, in this test, the effect of reducing the strain of the cabinet with the seismic isolator was up to 99%, in comparison to the case without the seismic isolator.

### 6.4. Response of Accelerations

To evaluate the effect of the seismic motion reduction by the seismic isolator, the peak acceleration responses of both the cases were compared, as shown in Figure 12. Figure 12a,b show the peak accelerations related to both the recorded earthquakes. In addition, the peak accelerations corresponding to the artificial earthquakes are illustrated in Figure 12c–e. For Case 1, wherein the base channel is fixed to the concrete foundation with post-installed anchors, in the case of the Gyeongju earthquake (Figure 12a), the peak acceleration response of the internal component power supply and relay was greater than that of the other locations. In particular, the peak acceleration response was 3.78 g, which is the largest in the relay. In addition, for the case of the Pohang earthquake (Figure 12b), the peak acceleration response was 2.22 g at the relay. In the case of the artificial earthquake in Figure 12c–e, the peak acceleration responses were also measured at the top of the panel's enclosure, which were 23.96 g, 24.14 g, and 35.81 g for EQ #1, #2, and #3, respectively. The peak acceleration response measured by the internal power supply and relay was also greater than 5 g for Case 1. In Case 2 of the RTU panel installed with non-overturn seismic isolator, however, the peak acceleration response in the Gyeongju earthquake, Pohang earthquake, and EQ #1 was approximately 0.2 g; the peak acceleration response up to EQ #2, the seismic design level, was less than 0.5 g. Even in EQ #3, the magnitude of the peak acceleration response was quite small compared to Case 1. Figure 13 showed the reduction ratio of the peak acceleration response of Case 1 and Case 2 by Equation (12). Here, $a_{max}$ is peak acceleration of each case. Except for EQ #3, the peak acceleration response of Case 2 in the horizontal direction was at least 85.3%, and up to 99.6% smaller than that of Case 1. In the vertical direction, except for the Pohang earthquake and EQ #3, the peak accelerations measured in Case 2 were approximately 79.7%–99.7% less than those of Case 1. Therefore, the non-overturn seismic isolator applied in this study can effectively control the vibration amplification of the structures during triaxial simultaneous input ground motions. The peak acceleration ratio in the vertical direction, however, was less than that in the horizontal direction.

$$R_a = \left(1 - \frac{a_{max,case1}}{a_{max,case2}}\right) \times 100 \tag{12}$$

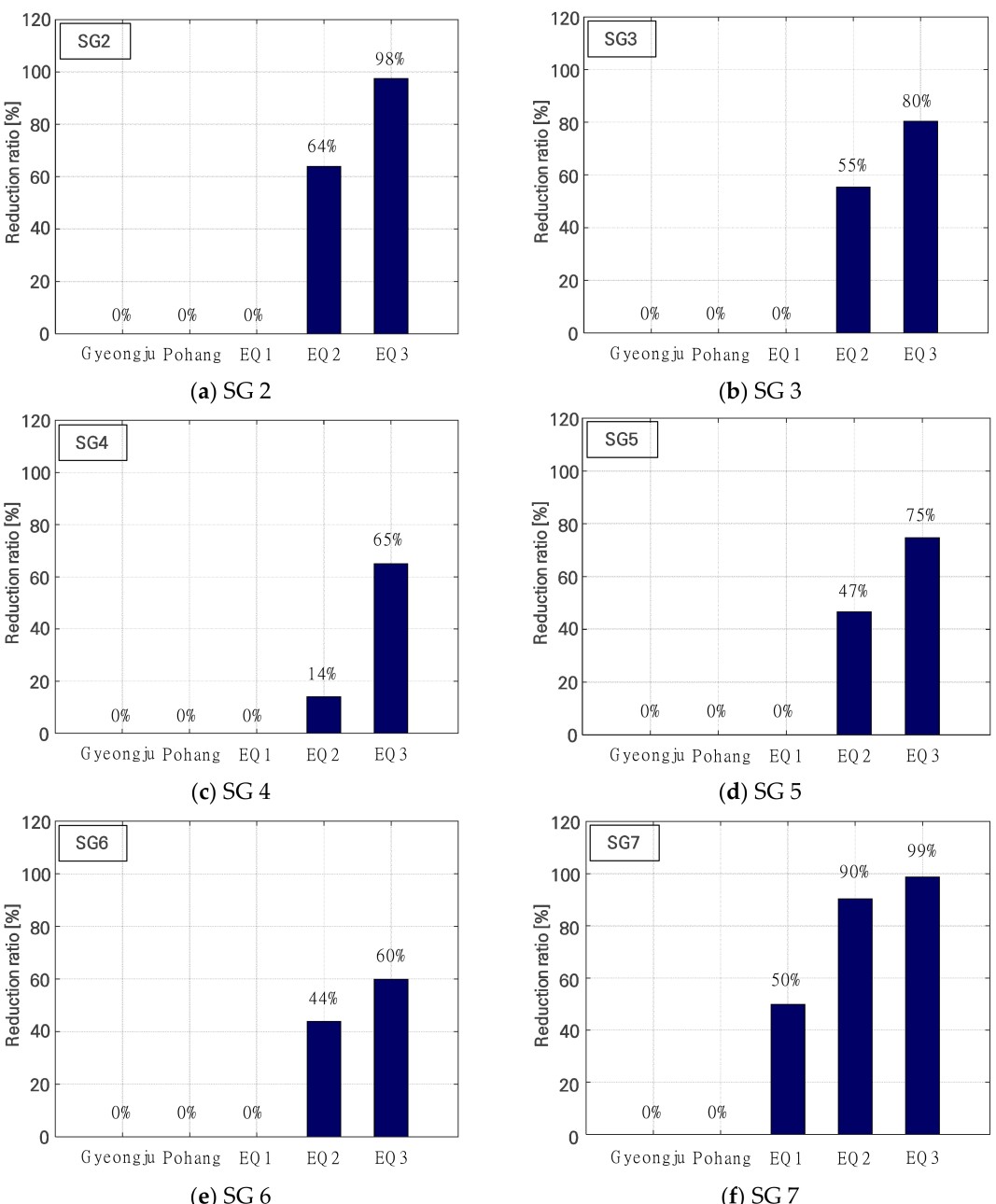

**Figure 11.** Permanent strain ratios (Case 2/Case 1).

### 6.5. Amplification of the Response Spectrum Acceleration

The amplification ratio of the response spectrum acceleration, $F_a(i)$, of the RTU system subjected to seismic ground motion can be expressed by Equation (13) [37].

$$F_a(i) = \frac{S_a(component)}{S_a(ground)} \tag{13}$$

where $S_a(ground)$ is the TRS measured at the top of the concrete foundation installed on the shaking table, and $S_a(component)$ is the TRS measured by the accelerometer attached to the RTU panel. TRS was analyzed as a 1/12 octave.

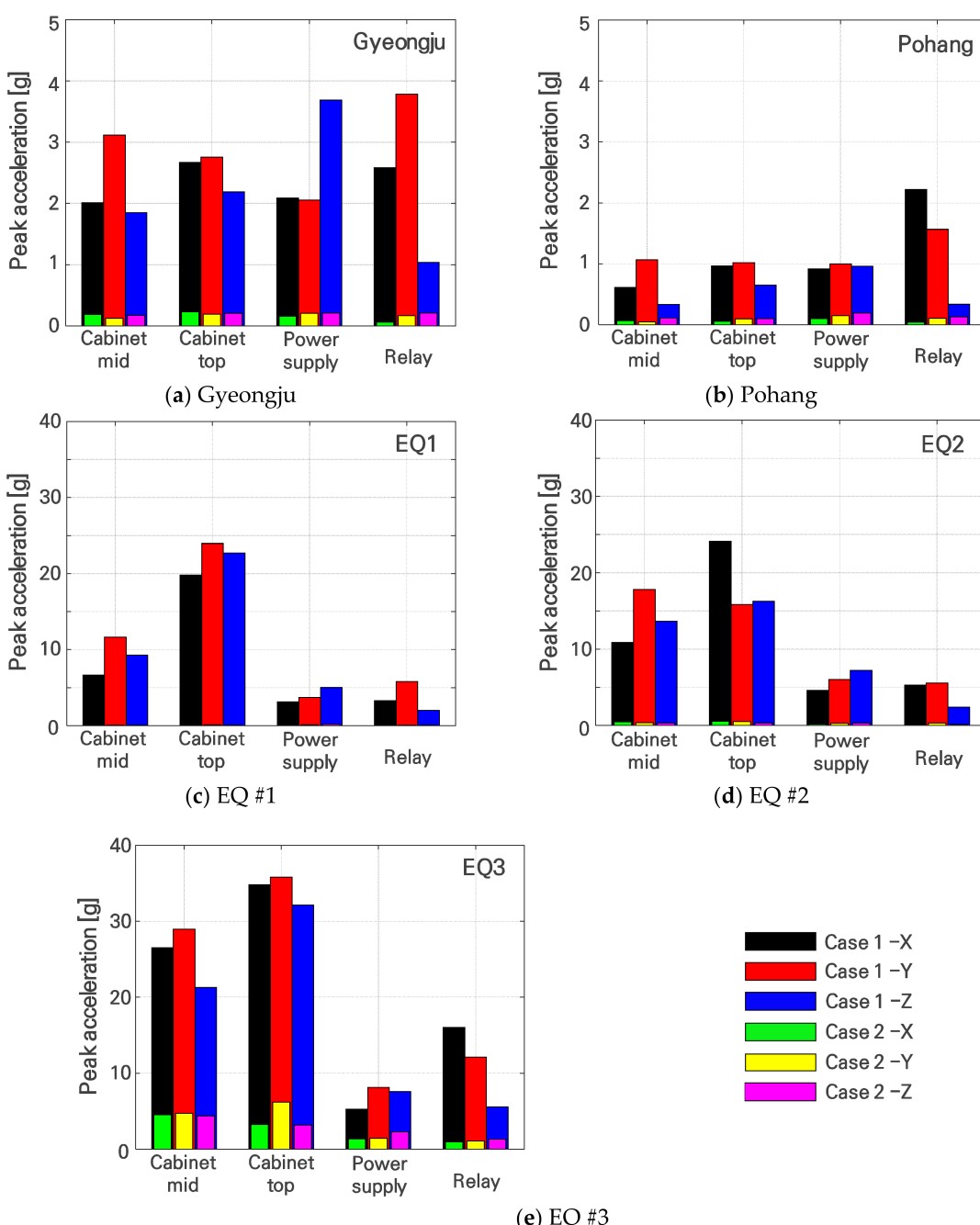

**Figure 12.** Comparison of peak acceleration responses.

Figures 14 and 15 demonstrate $F_a(i)$ for each measurement location for case 1, and 2, respectively. In both cases, the amplification in the vertical direction of the power supply was the largest; in addition, $F_a$ of the power supply for Case 1 was 65. However, for Case 2, the amplification ratio was efficiently controlled to 6 or less. For Case 1, in all earthquakes, excluding the Pohang earthquake, $F_a$ tended to increase rapidly in the frequency region above 30 Hz. The amplification ratio of the response may appear large in the high-frequency region owing to the vibration of the device, thumping of the door, and the impact caused by uplifting [12]. In addition, for Case 2, the peak $F_a$ value up to the EQ #2 test was maintained to less than 1 in both the horizontal and vertical directions. In the EQ #3 test, $F_a$ increased from the RTU panel's enclosure toward the high-frequency range. Notably, the ball and slider plate collided with the allowable limit displacement of the seismic isolator. Compared with Case 1, the value for Case 2 was less than 1/7. Table 11

lists the $F_a$ values corresponding to zero period acceleration (ZPA) associated with the maximum peak acceleration of the acceleration time history. ZPA is the acceleration level of the high-frequency, non-amplified portion of the response spectrum. This acceleration corresponds to the maximum (peak) acceleration of the time history used to derive the spectrum. In the IEEE std 693, which is the shaking table test method, the ZPA is assumed to be the acceleration at 33 Hz or greater [35]. In this study, because the input ground motion was identified within a range of 0–50 Hz, ZPA was considered to be 50 Hz. For Case 2, the peak $F_a$ values up to EQ #2 were less than 1. In particular, the $F_a$ of internal equipment to EQ #1 based on the building structure standards and EQ #2 related to the common application of seismic design standards was well controlled at 0.14–0.4. Figure 16 depicts the ratio of $F_a$ for cases 1 and 2 in Table 11. As shown in Figure 13, except for EQ #3, the ratio of the peak acceleration response in Case 2 was 85% or more in both horizontal directions, and 75% or more in the vertical direction. Consequently, the non-overturn seismic isolator installed in this study effectively controls the amplification of seismic ground motion within the seismic demand range.

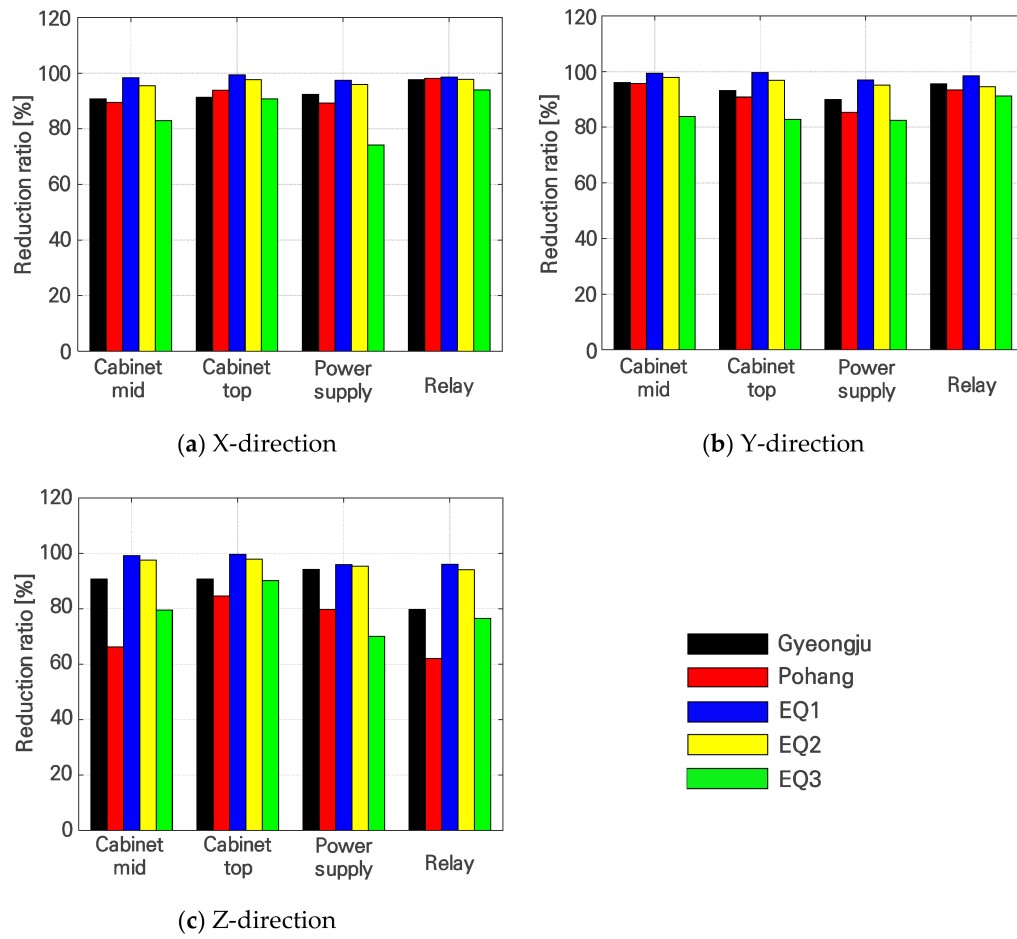

**Figure 13.** Ratio of peak accelerations (Case 2/Case 1).

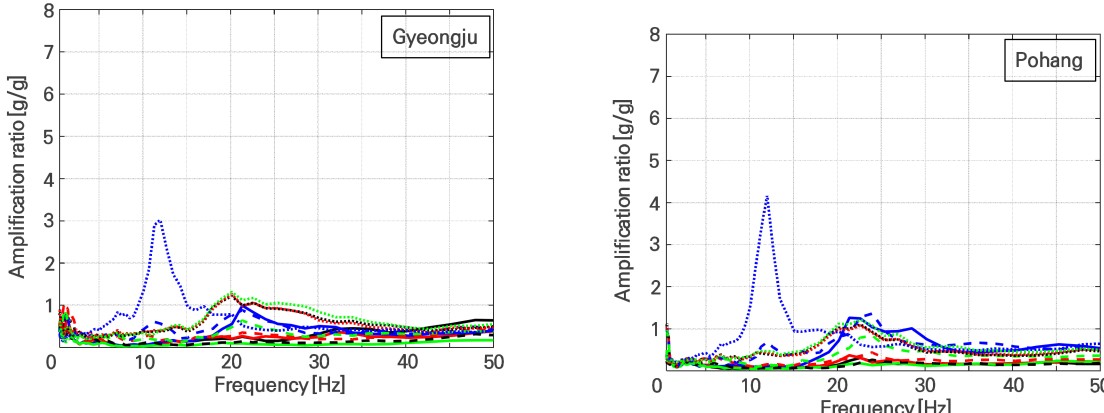

**Figure 14.** $F_a$ values for Case 1.

**Figure 15.** *Cont.*

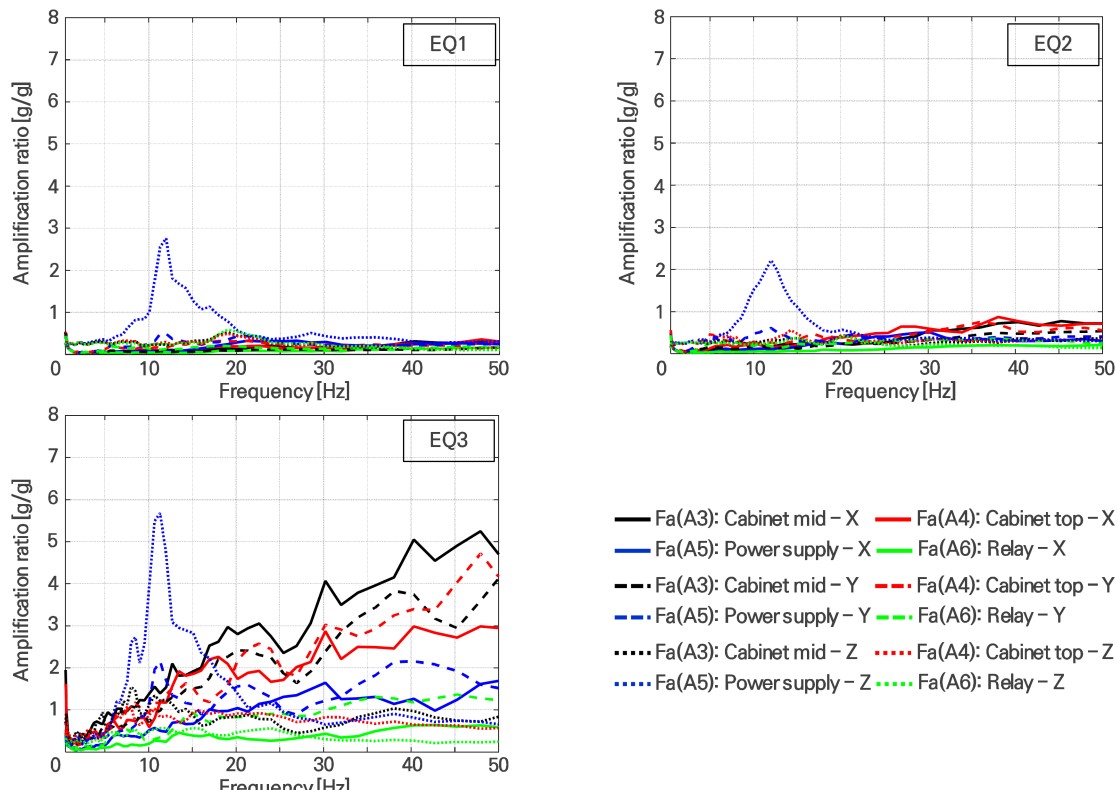

**Figure 15.** $F_a$ values for Case 2.

**Table 11.** Peak value of $F_a$ at ZPA of 50 Hz.

| Test No. (Input Motion) | Case | Direction | Sensor No. (Location) | | | | |
|---|---|---|---|---|---|---|---|
| | | | A2 (Cabinet Bottom) | A3 (Cabinet Mid) | A4 (Cabinet Top) | A5 (Power Supply) | A6 (Relay) |
| #1 (Gyeongju) | Case 1 (*g*) | X | 5.30 | 7.93 | 7.45 | 7.16 | 5.96 |
| | | Y | 7.08 | 8.52 | 8.69 | 8.36 | 10.44 |
| | | Z | 8.14 | 5.65 | 7.77 | 12.65 | 6.34 |
| | Case 2 (*g*) | X | 0.26 | 0.65 | 0.40 | 0.42 | 0.17 |
| | | Y | 0.43 | 0.28 | 0.42 | 0.40 | 0.36 |
| | | Z | 0.51 | 0.51 | 0.49 | 0.41 | 0.56 |
| | Ratio (Case 2/Case 1) (%) | X | 95.18 | 91.86 | 94.62 | 94.12 | 97.20 |
| | | Y | 93.94 | 96.66 | 95.22 | 95.25 | 96.59 |
| | | Z | 93.75 | 90.99 | 93.73 | 96.74 | 91.11 |
| #2 (Pohang) | Case 1 (*g*) | X | 2.86 | 3.96 | 3.03 | 5.42 | 3.39 |
| | | Y | 2.10 | 4.40 | 5.88 | 6.59 | 7.83 |
| | | Z | 2.69 | 2.12 | 3.21 | 4.31 | 2.04 |
| | Case 2 (*g*) | X | 0.21 | 0.16 | 0.22 | 0.52 | 0.22 |
| | | Y | 0.20 | 0.19 | 0.27 | 0.50 | 0.36 |
| | | Z | 0.46 | 0.45 | 0.49 | 0.65 | 0.48 |
| | Case 2/Case 1 | X | 92.81 | 95.94 | 92.87 | 90.47 | 93.54 |
| | | Y | 90.27 | 95.63 | 95.34 | 92.47 | 95.38 |
| | | Z | 82.95 | 78.68 | 84.68 | 84.84 | 76.29 |

**Table 11.** *Cont.*

| Test No. (Input Motion) | Case | Direction | Sensor No. (Location) | | | | |
|---|---|---|---|---|---|---|---|
| | | | A2 (Cabinet Bottom) | A3 (Cabinet Mid) | A4 (Cabinet Top) | A5 (Power Supply) | A6 (Relay) |
| #1 (EQ1) | Case 1 (*g*) | X | 21.17 | 9.04 | 24.91 | 8.44 | 5.06 |
| | | Y | 9.07 | 17.26 | 22.78 | 8.83 | 7.24 |
| | | Z | 7.34 | 5.12 | 12.18 | 6.12 | 3.69 |
| | Case 2 (*g*) | X | 0.15 | 0.32 | 0.29 | 0.23 | 0.17 |
| | | Y | 0.24 | 0.17 | 0.27 | 0.24 | 0.17 |
| | | Z | 0.11 | 0.12 | 0.16 | 0.32 | 0.11 |
| | Case 2/Case 1 | X | 99.30 | 96.49 | 98.84 | 97.26 | 96.60 |
| | | Y | 97.39 | 99.02 | 98.83 | 97.26 | 97.69 |
| | | Z | 98.48 | 97.72 | 98.70 | 94.84 | 96.98 |
| #2 (EQ2) | Case 1 (*g*) | X | 12.50 | 12.54 | 11.02 | 6.74 | 4.37 |
| | | Y | 10.29 | 21.57 | 18.38 | 9.05 | 8.50 |
| | | Z | 9.29 | 6.71 | 10.52 | 6.62 | 2.80 |
| | Case 2 (*g*) | X | 0.54 | 0.72 | 0.72 | 0.29 | 0.22 |
| | | Y | 0.60 | 0.54 | 0.53 | 0.40 | 0.25 |
| | | Z | 0.19 | 0.36 | 0.32 | 0.29 | 0.14 |
| | Case 2/Case 1 | X | 95.65 | 94.27 | 93.50 | 95.67 | 94.98 |
| | | Y | 94.17 | 97.48 | 97.13 | 95.54 | 97.06 |
| | | Z | 97.95 | 94.57 | 96.97 | 95.56 | 94.98 |
| #3 (EQ3) | Case 1 (*g*) | X | 34.72 | 36.70 | 26.76 | 5.53 | 12.22 |
| | | Y | 29.70 | 25.53 | 30.68 | 8.38 | 9.18 |
| | | Z | 17.65 | 9.48 | 19.12 | 7.29 | 5.26 |
| | Case 2 (*g*) | X | 1.74 | 4.49 | 2.93 | 1.72 | 0.57 |
| | | Y | 3.28 | 4.33 | 3.95 | 1.49 | 1.21 |
| | | Z | 0.41 | 0.88 | 0.57 | 0.64 | 0.25 |
| | Case 2/Case 1 | X | 95.00 | 87.78 | 89.05 | 68.82 | 95.33 |
| | | Y | 88.95 | 83.06 | 87.11 | 82.24 | 86.78 |
| | | Z | 97.66 | 90.68 | 97.02 | 91.27 | 95.26 |

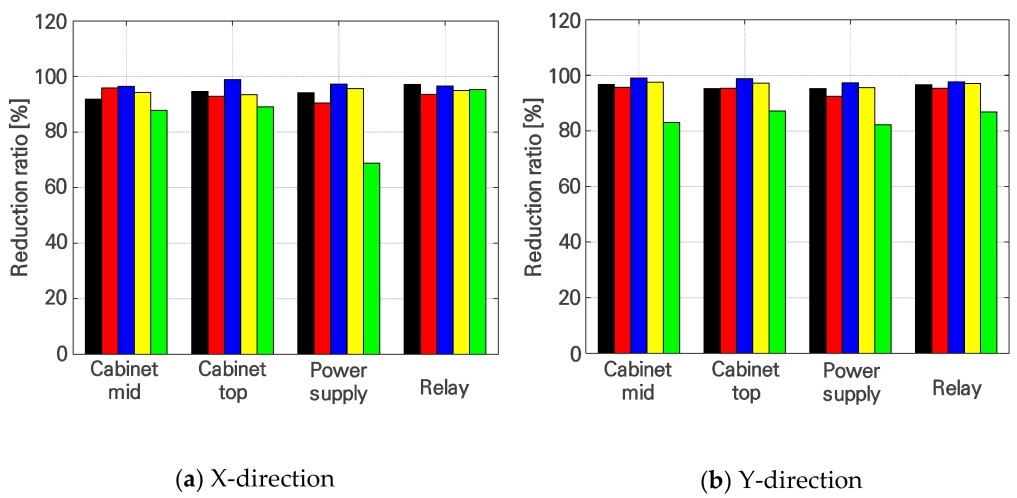

(**a**) X-direction          (**b**) Y-direction

**Figure 16.** *Cont.*

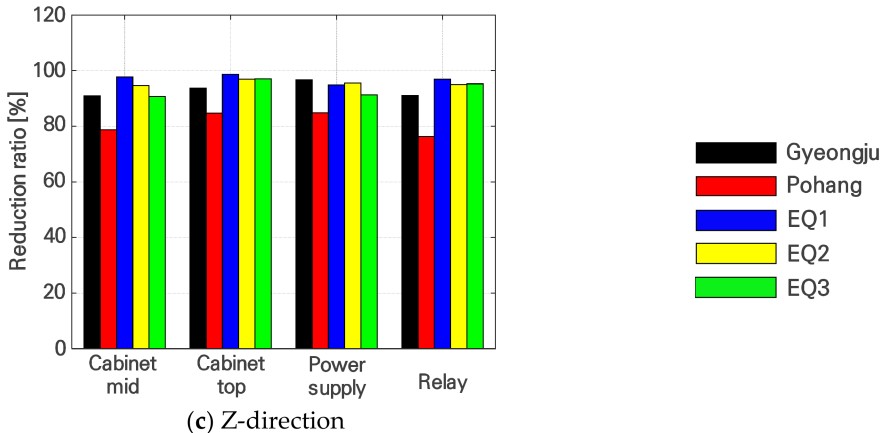

(**c**) Z-direction

**Figure 16.** Ratio of $F_a$ values (Case 2/Case 1).

## 7. Discussion and Conclusions

In this study, shaking table tests were conducted, and the observations from the experimental tests were compared under two different conditions: (a) the RTU panel fixed to the concrete slab and (b) the RTU panel installed with the non-overturn seismic isolator using an FPS. The shaking table test was conducted according to ICC-ES AC 156, and the seismic simulation test was performed with a three-axis simultaneous excitation test. In addition, the Gyeongju and Pohang earthquakes, as recorded ground motions, were considered by the input ground motions in this study. Artificial earthquakes were generated based on the common application of building seismic-resistant design standards and earthquake-resistant design standards. The exploratory study of dynamic characteristics of the RTU systems was as follows.

- For Case 1, wherein the RTU panel was fixed to the concrete foundation using a set anchor, the peak acceleration response of the internal components' power supply and relay was greater than that of other locations for the recorded earthquake test. In contrast, during the artificial earthquake tests, the peak acceleration response measured at the top of the panel enclosure was the highest.

- The acceleration response measured at the panel enclosure was significant owing to the impact caused by shaking or lifting of the door of the RTU system. In addition, such an impact can amplify vibrations in the high-frequency region.

- For the case of the RTU system installed with the non-overturn seismic isolator (Case 2), the peak acceleration response up to EQ #3, where the RTU panel was damaged in Case 1, was less than 70%, in comparison to that in Case 1.

- Moreover, the permanent deformation of the seismic isolator in the RTU panel was not observed around the lower plate bolts in the EQ #4 test. The amplification of the seismic force was investigated by the amplification ratio ($F_a$) of the TRS of the acceleration signal measured at the shaking table bottom and the locations of the sensors.

- In both cases 1 and 2, the acceleration in the vertical direction of the power supply was significantly amplified, and the amplification ratio ($F_a$) value of Case 1 rapidly increased in the frequency range above 30 Hz. Furthermore, the vibration of the device, shaking of the door, and uplifting in the high-frequency content affected the amplification ratio of the responses considerably.

- For Case 2, the peak values of $F_a$ up to the test of EQ #2 related to the level of the design earthquake was effectively controlled to be less than 1 in both the horizontal and vertical directions. The $F_a$ value of ZPA (50 Hz) corresponding to the peak value of the acceleration time history was evaluated.

- The $F_a$ value of critical internal equipment associated with the artificial earthquakes EQ #1 and #2 for Case 2 was 0.4 and 0.11, respectively. Therefore, the non-overturn seismic isolator applied to the lower part of the RTU panel effectively controlled the

internal equipment against artificial earthquakes (EQ #1 and #2), considering the seismic design standards.

- In addition, various damage modes, such as the fall of the wire mold, damage to the door lock, door opening, and damage to the anchor bolt, occurred in the RTU panel fixed using a set anchor on the concrete foundation; however, for the case of the RTU system installed with the non-overturn seismic isolator, the damage was observed only at the seismic isolator.

Furthermore, in the RTU panel fixed to the concrete foundation with a set anchor, cup-like deformation occurred around the bolt of the lower plate due to strong ground motion. Moreover, the relative displacement between the top and bottom of the RTU panel significantly increased owing to the seismic load acting in the left and right directions. The impact generated by the lift RTU panel colliding with the lower channel made the internal equipment respond, and the panel enclosure was capable of having a significant impact. However, the application of the seismic isolator can effectively protect the RTU panel from the impact and amplification of seismic forces. In particular, in this study, the non-overturn seismic isolator installed to achieve additional seismic resilience contained a spring that generated a certain level of restraining force in the vertical direction, effectively dealing with the amplification of the seismic force, although a friction pendulum type of the seismic isolator was applied in the horizontal direction. However, it showed that the test was conducted for a recorded earthquake with a PGA of 0.43 g or less and an artificial earthquake with a vertical component of 27% of a horizontal earthquake.

**Author Contributions:** Conceptualization, S.-W.K., B.-G.J., D.-W.Y. and W.-Y.J.; experimental tests, S.-W.K., B.-G.J., D.-W.Y. and B.-S.J.; methodology, S.-W.K., B.-G.J., D.-W.Y., W.-Y.J. and B.-S.J.; formal analysis, S.-W.K., B.-G.J. and D.-W.Y.; data curation, B.-G.J. and B.-S.J.; writing—original draft preparation, B.-G.J., W.-Y.J. and B.-S.J.; writing—review and editing. All authors have read and agreed to the published version of the manuscript.

**Funding:** This research was supported by Ministry of Land, Infrastructure and Transport of Korean government.

**Institutional Review Board Statement:** Not applicable.

**Informed Consent Statement:** Not applicable.

**Data Availability Statement:** Data are contained within the article.

**Acknowledgments:** This research was supported by a grant (21IFIP-B128598-05) from Industrial Facilities and Infrastructure Research Program (IFIP) funded by Ministry of Land, Infrastructure and Transport of Korean government.

**Conflicts of Interest:** The authors declare no conflict of interest.

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
