# Peer review of "Seismic Experimental Assessment of Remote Terminal Unit System with Friction Pendulum under Triaxial Shake Table Tests"

_metals, doi:10.3390/met11091428_

Round 1

Reviewer 1 Report

The paper entitled “Experimental Assessment of Vibration Characteristics of Seismic Isolated Remote Terminal Unit System with Friction Pendulum under Triaxial Shake Table Test”, presents an analysis of the dynamic characteristics of a remote terminal unit with a friction pendulum as the electrical cabinet in a critical facility, using a triaxial shaking table test. In addition, the mitigation of the seismic characteristics of the cabinet system is evaluated in comparison to the non-seismic isolated cabinet system.

COMMENTS

The Abstract of the paper MUST be improved to better represent the paper. Lines 13-22 are introductory and wordy and only lines 23-26 directly refer to the paper. Some basic conclusions could be added.

Equation (5) should be corrected (c at left should be deleted).

Equation (6) should be corrected (c at left should be deleted).

I suggest to authors to add/modify as follows

ADD        7. Discussion

MODIFY 8. Conclusions

Some parts of the “7. Conclusion” should go to “7. Discussion” and the “8. Conclusions” could be more focused; bullets (⦁) could clarify results.

Author Response

Thank you for the valuable review comments. Please see the attachment. 

Reviewer 2 Report

The paper presents the results of a series of dynamic characteristics of a of seismic non-isolated versus isolated remote terminal unit (electrical cabinet). The isolated system considers a base isolation through a Friction Pendulum isolator. The Dynamic loading was applied by a tri-axial shaking table, by considering real and artificial accelerograms.

The article clearly presents the cases consdered, as well as the obtained results. However, the authors could improve the clearness of the article by considering the following observations:

  1. The title of the article reffer only to the vibration characteristics, while the results resent the seismic response of such devices. Besides, the title is too long in present form.
  2.  The geometrical dimensions of RTU 2-1 and respectively 3-1 as described in Table 2 are identical.
  3.  Define the TPF acronym in the body-text
  4.  The authors should explain the dimensions of anchoring bolts used in the two analysed cases: M8 for the classic RTU and M30 for the isolated RTU pannel
  5.  Perhaps it would be instructive to complete the figure 5 with the artificial accelerogram considered in the dynamic analyses. Thus, one can have a direct comparison of the recorded and artificial motions considered.
  6.  In Table 4 the authors mention the 5% damping of the system. This value should be argumented, as the usual value of the steel systems is of 2%.
  7.  Part of the text is mismatched in lines 366-379
  8.  For  good understanding, the zero period acceleration (ZPA) should be explained. Further, ZPA cannot be measured in Hz, as given in lines 575-577.
  9.  for a better view of results it is advisable to express the results of the strain gauges in microstrains.
  10.  The response of accelerations is well explained in paragraph 6.4 and this proves the efficiency of the isolation system. However, it would be instructive to know the amplification given by the RTU (original and isolated) with regard to the ZTA.

Author Response

(The authors gave the same response as above.)

Round 2

Reviewer 1 Report

The revised paper can now be accepted for publication.

Author Response

“Seismic Experimental Assessment of Remote Terminal Unit System with Friction Pendulum under Triaxial Shake Table Tests

 (Manuscript # metals-1322265)

We thank the reviewers for their valuable review. We have revised the manuscript as requested. Specific steps taken to address each comment are described below and the changes directly made to the manuscript were highlighted in yellow color. We hope that it may be now suitable for publication.

Reviewer 1 Comments:

The revised paper can now be accepted for publication.

Authors Reply:

We thank the reviewer for a careful reading.

Reviewer 2 Report

The authors substantially improved the manuscript according to the review comments. However, for the clarity of the final manuscript, the following changes are suggested:

  1. (original comment 4): The difference in fixing between M8, M12 and M30 is huge. Give the readers some design values that justify this rough change in the supporting.
  2. (original comment 7): Just as comment: the 5% damping seems to be too high for this system 
  3. (original comment 10): Adjust to microstrains also the values given in table 8
  4. (original comment 11): It is suggested to explain if there is ay difference between Sa(ground) and ZPA. Normally there should be no important differences between the two.

Author Response

“Seismic Experimental Assessment of Remote Terminal Unit System with Friction Pendulum under Triaxial Shake Table Tests

 (Manuscript # metals-1322265)

We thank the reviewers for their valuable review. We have revised the manuscript as requested. Specific steps taken to address each comment are described below and the changes directly made to the manuscript were highlighted in yellow color. We hope that it may be now suitable for publication.

Reviewer 2 Comments:

The authors substantially improved the manuscript according to the review comments. However, for the clarity of the final manuscript, the following changes are suggested:

  1. (original comment 4): The difference in fixing between M8, M12 and M30 is huge. Give the readers some design values that justify this rough change in the supporting.

Authors Reply:

We thank the reviewer for a careful reading. In order to address this comment, we have added the design description of the anchors at the end of the section 3.

  1. (original comment 7): Just as comment: the 5% damping seems to be too high for this system 

Authors Reply:

We thank the reviewer for a careful reading. The RRS used in this paper refers to ICC-ES AC 156, and ICC-ES AC 156 suggested 5% as a damping ratio.

  1.  (original comment 10): Adjust to microstrains also the values given in table 8

Authors Reply:

We thank the reviewer for a careful reading. It has been modified in the table 8.

  1. (original comment 11): It is suggested to explain if there is ay difference between Sa(ground) and ZPA. Normally there should be no important differences between the two.

Authors Reply:

We thank the reviewer for a careful reading. Here, Sa(component) and Sa(ground) are the spectral accelerations at each frequency in the test response spectrum (TRS) estimated as 1/12 octave using the acceleration response of the measured position, and ZPA is the zero period acceleration as mentioned in this paper. In this paper, the value with respect to ZPA was applied at 50 Hz point.